# Mercury isotope evidence for Arctic summertime re-emission of mercury from the cryosphere

Beatriz Ferreira Araujo[1,16], Stefan Osterwalder[2,3,16] ✉, Natalie Szponar[4,16], Domenica Lee[4], Mariia V. Petrova[5], Jakob Boyd Pernov[6,7], Shaddy Ahmed[3], Lars-Eric Heimbürger-Boavida[5], Laure Laffont[1], Roman Teisserenc[8], Nikita Tananaev[9,10], Claus Nordstrom[6], Olivier Magand[3], Geoff Stupple[11], Henrik Skov[6], Alexandra Steffen[11], Bridget Bergquist[4], Katrine Aspmo Pfaffhuber[12], Jennie L. Thomas[3], Simon Scheper[13,14], Tuukka Petäjä[15], Aurélien Dommergue[3] & Jeroen E. Sonke[1] ✉

During Arctic springtime, halogen radicals oxidize atmospheric elemental mercury ($Hg^0$), which deposits to the cryosphere. This is followed by a summertime atmospheric $Hg^0$ peak that is thought to result mostly from terrestrial Hg inputs to the Arctic Ocean, followed by photoreduction and emission to air. The large terrestrial Hg contribution to the Arctic Ocean and global atmosphere has raised concern over the potential release of permafrost Hg, via rivers and coastal erosion, with Arctic warming. Here we investigate Hg isotope variability of Arctic atmospheric, marine, and terrestrial Hg. We observe highly characteristic Hg isotope signatures during the summertime peak that reflect re-emission of Hg deposited to the cryosphere during spring. Air mass back trajectories support a cryospheric Hg emission source but no major terrestrial source. This implies that terrestrial Hg inputs to the Arctic Ocean remain in the marine ecosystem, without substantial loss to the global atmosphere, but with possible effects on food webs.

Mercury (Hg) is a global pollutant that bioaccumulates in aquatic food webs and leads to health issues for humans and wildlife[1,2]. Human activities have greatly increased Hg inputs to the global environment mainly through mining and industrial activities[3]. Anthropogenic Hg emissions to the atmosphere are estimated to be around 2500 Mg y$^{-1}$[4], exceeding natural Hg emissions of 340 Mg y$^{-1}$ by sevenfold[5]. Gaseous elemental $Hg^0$ emissions disperse globally, due to the relatively long atmospheric $Hg^0$ lifetime (~5 months[6]), and can reach the Arctic by long-range atmospheric transport[7]. Deposition of atmospheric Hg to Arctic marine ecosystems, microbial conversion to methylmercury[8,9]

[1]Géosciences Environnement Toulouse, CNRS, IRD, Université de Toulouse, Toulouse, France. [2]Institute of Agricultural Sciences, ETH Zurich, Zurich, Switzerland. [3]University Grenoble Alpes, CNRS, IRD, Grenoble INP, IGE, Grenoble, France. [4]Department of Earth Sciences, University of Toronto, Toronto, ON, Canada. [5]CNRS/INSU, Aix Marseille Université, Université de Toulon, IRD, Mediterranean Institute of Oceanography, Marseille, France. [6]Department of Environmental Science, iClimate, Aarhus University, Roskilde, Denmark. [7]Extreme Environments Research Laboratory, École Polytechnique fédérale de Lausanne, Sion, Switzerland. [8]UMR5245 CNRS/UPS/INPT, Laboratoire Écologie Fonctionnelle et Environnement, Toulouse, France. [9]Melnikov Permafrost Institute, Siberian Branch, Russian Academy of Sciences, Yakutsk, Russia. [10]Institute of Natural Sciences, North-Eastern Federal University, Yakutsk, Russia. [11]Air Quality Research Division, Environment and Climate Change Canada, Toronto, ON, Canada. [12]Norwegian Institute for Air Research, Kjeller, Norway. [13]Dr. Simon Scheper—Research | Consulting | Teaching, Dähre, Germany. [14]Environmental Geosciences, University of Basel, Basel, Switzerland. [15]Institute for Atmospheric and Earth System Research, University of Helsinki, Helsinki, Finland. [16]These authors contributed equally: Beatriz Ferreira Araujo, Stefan Osterwalder, Natalie Szponar. ✉e-mail: stefan.osterwalder@usys.ethz.ch; jeroen.sonke@get.omp.eu

and subsequent biomagnification along marine food webs expose indigenous populations to Hg through their traditional diet of high trophic level seafood[10].

Atmospheric Hg is distributed primarily in three different chemical forms: gaseous elemental $Hg^0$, gaseous oxidized $Hg^{II}$ and particulate bound $Hg^{II}$. Redox reactions between $Hg^0$ and $Hg^{II}$ in the atmosphere are mostly photochemically driven[6,11], and both forms can deposit to terrestrial and marine ecosystems[12]. Many efforts have been made to characterize and understand the atmospheric transport, delivery and fate of Hg to the Arctic[10]. In 1998 Schroeder and collaborators reported on atmospheric mercury depletion events (AMDEs) observed in springtime in the Arctic at Alert, Canada[13]. AMDEs are driven by sea-salt derived reactive halogen oxidants in the atmosphere following polar sunrise[14–17]. During AMDEs, $Hg^0$ is near-quantitatively removed from the atmosphere by oxidation to $Hg^{II}$ forms which subsequently deposit rapidly to snow (on ice and on land) or sea ice[18–20]. Oxidized $Hg^{II}$ species deposited during AMDEs can undergo photoreduction in snow where a large part (on average ~80%, observed at coastal sites[10]) is re-emitted as $Hg^0$ to the atmosphere. The re-emitted Hg fraction is lower from snow on sea ice than from snow on coastal land[21], due to the higher marine-derived concentrations of $Cl^-$ which inhibit $Hg^{II}$ photoreduction[22]. The integrated summertime rebound in $Hg^0$ at Alert represents 62% of the springtime drop in $Hg^0$ over the period 1995–2002[20]. Thus, a significant fraction of AMDE deposited Hg therefore remains in the snowpack and runs off with snow melt to impact freshwater and marine ecosystems[23,24].

The springtime atmospheric $Hg^0$ depletion (mean of 1.35 ng m$^{-3}$ for April–May, 2000–2009 period) observed at different monitoring stations across the Arctic including Alert, Villum, Zeppelin, Utqiagvik, and Amderma is generally followed by a summertime $Hg^0$ maximum of 1.80 ng m$^{-3}$ (July mean[25–27]). This unique Arctic $Hg^0$ seasonality suggests net Hg deposition in spring, followed by net Hg emission during summer[28]. The origin of the summertime $Hg^0$ peak is less well understood, and has been attributed to AMDE re-emissions[14], evasion from the AO[29], and long-range transport of Asian air[27]. In 2012 the GEOS-Chem atmospheric Hg chemistry and transport model was used to assess Arctic $Hg^0$ seasonality[25]. Using a number of sensitivity runs, it was found that neither cryosphere and ocean re-emissions, nor transport from mid-latitudes, could explain the summertime $Hg^0$ maximum. The authors suggested that the missing source was 95 Mg y$^{-1}$ of terrestrial Hg inputs to the AO from rivers and coastal erosion. In the model a large fraction of this terrestrial Hg is photoreduced in the surface AO and emitted to the atmosphere as $Hg^0$ during the early summer (corresponding to the onset of the sea ice melt season) resulting in the Arctic summertime $Hg^0$ maximum.

Subsequent model improvement refined this number to 62–97 Mg y$^{-1}$, divided between rivers (46–50 Mg y$^{-1}$)[28,30] and coastal erosion (16–47 Mg y$^{-1}$)[30–32]. Pan-Arctic seasonal river Hg observations have confirmed a large river contribution of 41 ± 4 Mg y$^{-1}$, delivered mostly during the spring flood in May-June[33,34]. The coastal erosion Hg flux remains uncertain, largely due to variation in the assumed glacial sediment Hg concentration and will be updated in this work.

In order to reproduce Arctic atmospheric $Hg^0$ seasonality, coupled Arctic air-sea Hg models require an unusually large fraction (80%) of terrestrial Hg inputs to be photoreduced in the AO compared to 8% in other ocean basins[30,34]. In addition, coastal erosion inputs lag river inputs by one and a half months, peaking late August and September, when sea ice cover is minimal and wave action on coast lines maximal[35]. These two caveats put into question the role of terrestrial Hg in driving the summertime atmospheric $Hg^0$ peak. Potential re-emission of terrestrial Hg to air from the surface AO has led to concerns that the large permafrost soil Hg pool (72,000 Mg in the upper 30 cm[36]) will be released to the global atmosphere with Arctic warming[37,38]. On the contrary, if terrestrial Hg is predominantly buried with sediments over the large AO shelf, then the predicted global atmospheric impact from river Hg and AO coastal erosion would be less. However, enhanced burial of terrestrial Hg in AO shelf sediments could possibly lead to increased in situ production of MeHg that can impact both benthic and pelagic marine food webs. As there is no observational evidence for re-emission of terrestrial Hg from the AO, its fate remains uncertain as well as its contribution to the higher summer $Hg^0$ in the Arctic atmosphere.

In this study, we explore the seasonal variability of Hg stable isotope signatures of atmospheric $Hg^0$ at Alert (ALT), Villum (VRS) and Zeppelin (ZEP) research stations (Fig. 1) to assess if the origin of the summertime $Hg^0$ maximum can be understood. Mass-dependent ($\delta^{202}Hg$) and mass-independent Hg isotope signatures ($\Delta^{199}Hg$, $\Delta^{200}Hg$, $\Delta^{201}Hg$, $\Delta^{204}Hg$) provide a wealth of information on Hg sources and transformations[39,40]. Previous Hg isotope studies in the Arctic have shown uniquely large $\Delta^{199}Hg$, $\Delta^{201}Hg$ fractionation during AMDE snow Hg re-emission[41], $\Delta^{200}Hg$ evidence for important tundra vegetation and soil $Hg^0$ uptake[42,43], a dominant atmospheric $Hg^0$ source in snowmelt Hg runoff[23], and biota $\Delta^{199}Hg$ variability controlled by sea ice[44,45]. We compliment new seasonal atmospheric $Hg^0$ isotope observations with additional isotope data on atmospheric $Hg^{II}$, snow $Hg^{II}$, Yenisei River dissolved $Hg^{II}$, and surface AO particulate $Hg^{II}$. The ensemble of new Hg isotope observations suggests that the Arctic summertime atmospheric $Hg^0$ maximum is caused by AMDE re-emissions and not by emission of terrestrial Hg inputs to the AO via river run-off and coastal erosion.

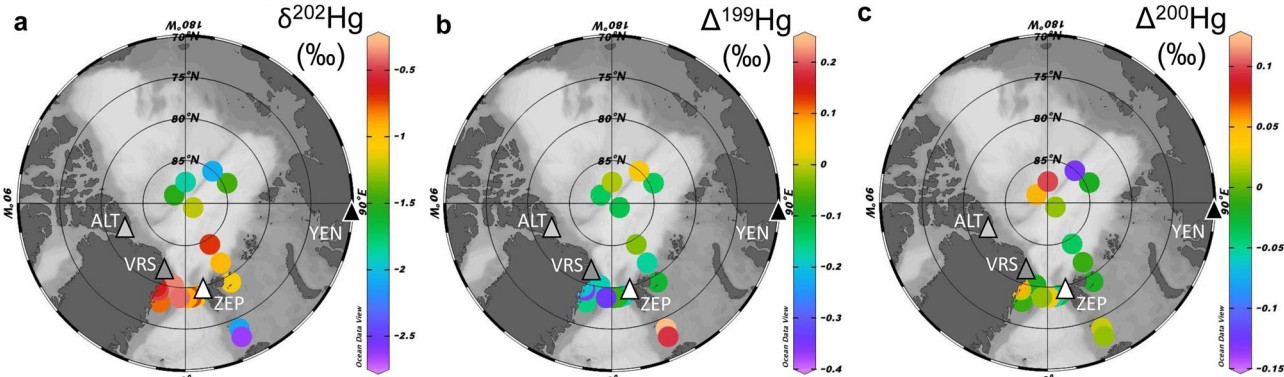

**Fig. 1 | Arctic sampling locations and marine Hg isotope variability.** Location of the three atmospheric research stations Alert (83°N, ALT), Villum (82°N, VRS) and Zeppelin (79°N, ZEP), and the city of Igarka, where the Yenisei River (YEN) was sampled (triangle symbols). Also shown are the surface Arctic Ocean (**a**) particulate Hg $\delta^{202}$Hg, (**b**) particulate Hg $\Delta^{199}$Hg and (**c**) particulate Hg $\Delta^{200}$Hg signatures (pHg, in ‰, round symbols) from the central basin and Barents Sea (this study) and from Fram Strait[60]. The figure was created using Ocean Data View[80] with permission from Alfred Wegener Institute, Helmholtz Center for Polar and Marine Research (AWI) (https://odv.awi.de/).

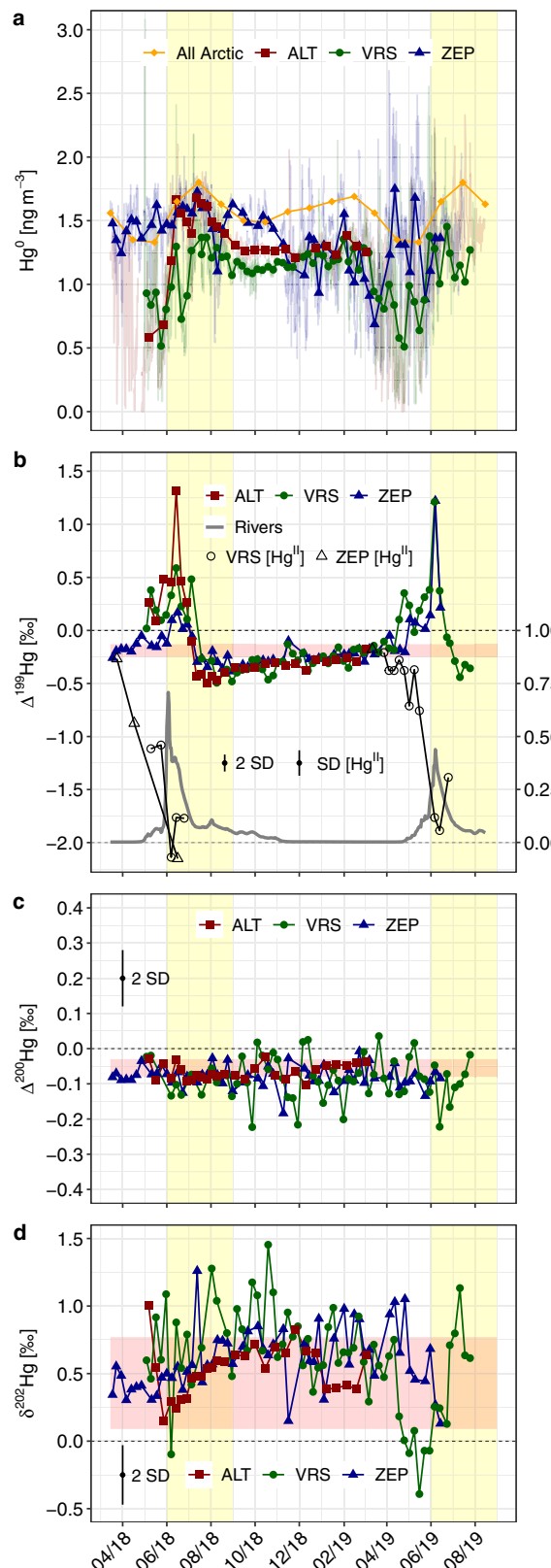

**Fig. 2 | Atmospheric Hg and Hg isotope seasonality in the Arctic.** Time series of (**a**) Hg⁰ concentrations, **b** Hg⁰ and HgᴵᴵΔ¹⁹⁹Hg, **c** Hg⁰ Δ²⁰⁰Hg and **d** Hg⁰ δ²⁰²Hg at Alert (ALT), Villum (VRS) and Zeppelin (ZEP) research stations. The panel a also shows the monthly mean long-term (2000–2009) Arctic Hg⁰ seasonality for ALT, ZEP and Amderma (All Arctic) from[25], with the summertime Hg⁰ maximum highlighted by the yellow shaded areas in all panels (vertical bars). The panel b also shows the predicted daily Eurasian river total Hg flux to the Arctic Ocean, based on[34]. Horizontal shaded areas (red) are the interquartile range of NH remote atmospheric Hg⁰ isotope variability. Dotted horizontal black lines indicate "0 per mil". The error bars in **b**, **c**, **d** represent the analytical precision determined as SD (Hgᴵᴵ) and 2 SD (Hg⁰), respectively from multiple measurements of an in-house standard.

mean of 1.51 ng m⁻³ for the year 2014 observed at 13 GMOS sites[46], and lower than at ZEP (1.50 ± 0.13 ng m⁻³), VRS (1.40 ± 0.21 ng m⁻³), and ALT (1.40 ± 0.21 ng m⁻³) from 2011 to 2015[47]. The low 2018–2019 Arctic Hg⁰ levels are compatible with continued declines in N-American and W-European Hg⁰ of −2% y⁻¹ observed between 1990 and 2012, but contrast with stable Arctic Hg⁰ levels over the same period[48]. The global decline in atmospheric Hg⁰ has been attributed to decreases in Hg emissions from oceans[49,50] and to decreases in anthropogenic emissions[48]. The lower Hg⁰ levels we observe for 2018–2019 therefore suggest that, despite a decade-long delay, the Arctic atmospheric Hg now follows the globally observed decline.

AMDEs are defined here as Hg⁰ levels below the 5th percentile of weekly mean Hg⁰, which are 0.92, 0.66 and 0.81 ng m⁻³ at ZEP, VRS and ALT, respectively. ZEP, at 474 m a.s.l., registered AMDEs during only four, short <12 h periods in April–May 2018, two times in Aug 2018, and on numerous prolonged occasions between November 2018 and May 2019. The ZEP polar winter dynamics are unusual, and did not occur over the 2000–2009 period[51]; they will not be further discussed here as they are beyond the scope of study. VRS, at 24 m a.s.l. in the planetary boundary layer, registered persistent AMDE events from April-June 2018, on three short occasions in Jan-Feb 2019, and again persistently from mid-March to May 2019. Numerous AMDEs at all sites were followed by short-lived increases in Hg⁰ above the 95th percentile of the data variability (Fig. 2a). These post-AMDE rebounds in Hg⁰ have been previously interpreted as photochemical Hg⁰ re-emission from snow Hgᴵᴵ [20,52,53]. Mean weekly Hg⁰ concentrations at ZEP and VRS show minor increases and a more pronounced rebound at ALT during July-August 2018 and June 2019 (Fig. 2a). Historically, the frequency of AMDEs, the springtime Hg⁰ minima and the summertime Hg⁰ maximum is strongest at ALT > VRS > Amderma (AMD) > ZEP and we observe the same trend here.

Weekly median atmospheric Hgᴵᴵ concentrations (collected on CEMs at VRS and ZEP, and with a Tekran® 1130–1135 system at ALT) during AMDE weeks were higher at VRS, 243 pg m⁻³ (IQR 82 to 412) and ALT, 118 ± 46 pg m⁻³ than at ZEP, 38 pg m⁻³ (IQR 21 to 53) due to the frequent arrival of air masses at ZEP from the free troposphere. Hg concentrations in 2019 snow samples from Ny-Ålesund ranged from 0.14 to 21.3 (5.2 ± 7.0) ng L⁻¹, which is similar to previous observations in the area[54,55], but lower than snow Hg levels up to 373 ng L⁻¹ during AMDE events[53].

### Hg⁰ stable isotopes and air mass back trajectories

Figures 2b, c, d and 3–5 show the Hg⁰ stable isotope variability of the dataset; NH background Hg⁰ isotope observations are indicated in Fig. 2b–d as horizontal shaded red bands, representing the IQR of published data. Hg⁰ at ZEP, VRS and ALT show overall similar positive median δ²⁰²Hg of 0.57‰ (IQR 0.45 to 0.75), 0.66‰ (IQR 0.47 to 0.85), 0.54‰ (IQR 0.39 to 0.64) and negative median Δ¹⁹⁹Hg of −0.20‰ (IQR −0.28 to −0.08), −0.21‰ (IQR −0.31 to 0.10), and −0.31‰ (IQR −0.15 to −0.38) respectively (Figs. 2b–d and 4). Δ¹⁹⁹Hg variability ranged from −0.50 to 1.32‰ at all three sites, which is uniquely larger than global Hg⁰ observations elsewhere (Figs. 2b and 4). Δ²⁰⁰Hg of Hg⁰ was

## Results and discussion

### Atmospheric Hg⁰ and Hgᴵᴵ, and snow THg concentrations

Average Hg⁰ concentrations at ZEP, VRS and ALT stations during the 2018–2019 campaigns were 1.35 ± 0.24 (mean ± 1 SD; range 0.69–1.75) ng m⁻³, 1.11 ± 0.19 (0.51–1.45) ng m⁻³, and 1.33 ± 0.25 (0.58–1.68) ng m⁻³ respectively (Fig. 2a and Supplementary Data File). These concentrations are lower than the northern hemisphere (NH)

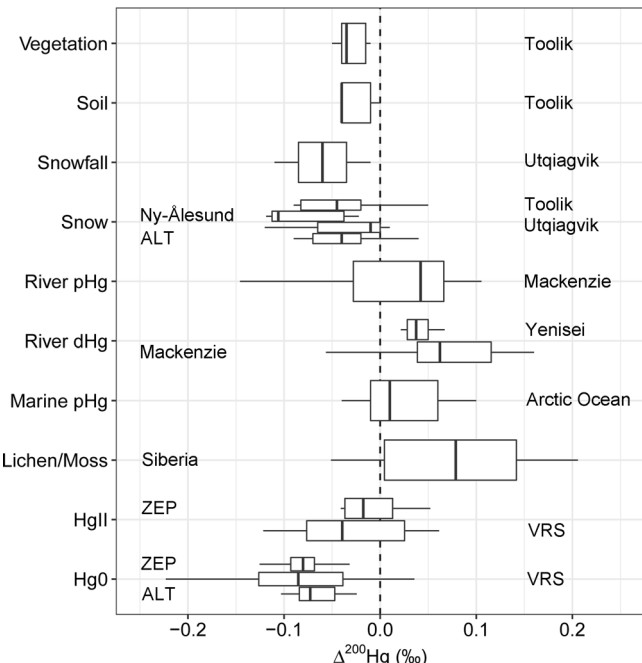

**Fig. 3 | Comparison of the $\Delta^{200}$Hg isotope signature of different Arctic matrices.** Toolik vegetation, soil, snow from[42,43], Utqiagvik Snowfall and snow from[41], Siberian tundra vegetation (lichen/moss) from[81]. The middle line in the box represents the median. The box limits indicate the upper and lower quartiles. The whiskers represent 1.5 times the interquartile range.

persistently negative, and did not vary at the three sites with median values of −0.08‰ (IQR −0.10 to −0.07), −0.09‰ (IQR −0.13 to −0.03), and −0.07‰ (IQR −0.09 to −0.05) at ZEP, VRS and ALT respectively (Fig. 2c). Limited observations were made for Hg⁰ $\Delta^{204}$Hg in ZEP (2019 only) and ALT samples, and show median $\Delta^{204}$Hg of 0.16‰ (IQR 0.12 to 0.19) and 0.12‰ (IQR 0.10 to 0.14) respectively. Overall, we report a $\Delta^{200}$Hg/$\Delta^{204}$Hg slope of −0.45 (Supplementary Fig. 1), which is in agreement with published atmospheric Hg data[40]. Our Hg⁰ isotope observations broadly agree with the limited observations from Utqiagvik (formerly Barrow; $n = 2$[41]), and ALT ($n = 2$[56]), except for the case of $\Delta^{199}$Hg at Utqiagvik, discussed below.

Figure 6 shows 10-day boundary layer air mass provenance maps for the months June, July and August at the three sites during both years of observation. June, July and August capture the start, peak and descent of the summertime atmospheric Hg⁰ maximum. Air mass provenance (hours per km²) is regional and does not originate over the Siberian shelf where the majority (88%[34]) of Arctic river and coastal erosion Hg inputs to the AO occur. Based on the HYSPLIT mixed layer depth (i.e., boundary layer height) and 10-day trajectory altitude profiles, we calculate that on average the air sampled at the three sites spent 21% of time in the boundary layer and 79% in the lower free troposphere (Fig. 5). The June air mass provenance within the boundary layer is associated by 62% with sea ice in the Lincoln Sea and snow-covered coastal land of Ellesmere Island and N-Greenland. The July Hg⁰ maximum and August descent remain associated with boundary layer sea ice and snow-covered land by 39%, but with an increasing proportion of boundary layer trajectories from Fram Strait, Nares Strait and Baffin Bay open waters (51%). Previous studies on the Arctic summertime Hg⁰ peak[20,25,30,34,57] did not assess air mass origins. Seasonal air mass provenance was determined using 10-day HYSPLIT back trajectories in a pan-Arctic aerosol study[58], finding similar lack of origins over Siberian coastal waters for ALT and VRS. A study on long-term Hg⁰ observations at VRS used 5-day back trajectories, finding no strong seasonal correlations between Hg⁰ and time spent over sea ice, open water, land or snow[26]. The study did not, however, produce

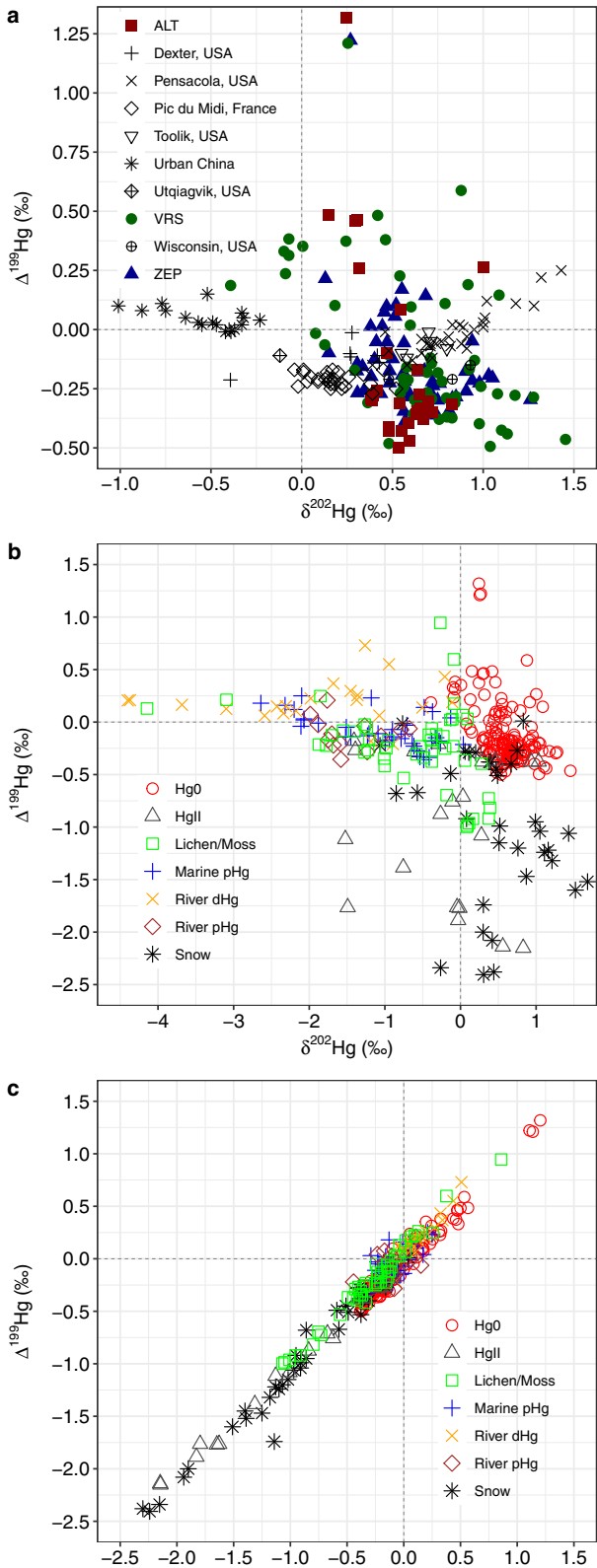

**Fig. 4 | Stable isotope signatures of Hg in the global atmosphere and the Arctic environment.** Mass-dependent ($\delta^{202}$Hg) and mass-independent ($\Delta^{199}$Hg) Hg⁰ isotope signatures (**a**) at different global locations and **b** for different Arctic pools. **c** Mass-independent ($\Delta^{201}$Hg) and mass-independent ($\Delta^{199}$Hg) isotope signatures for different Arctic pools. Arctic Hg⁰ $\Delta^{199}$Hg variability (−0.50 to 1.32‰) is much larger than the NH background $\Delta^{199}$Hg at remote sites. Reported literature data is summarized in the Supplementary Text.

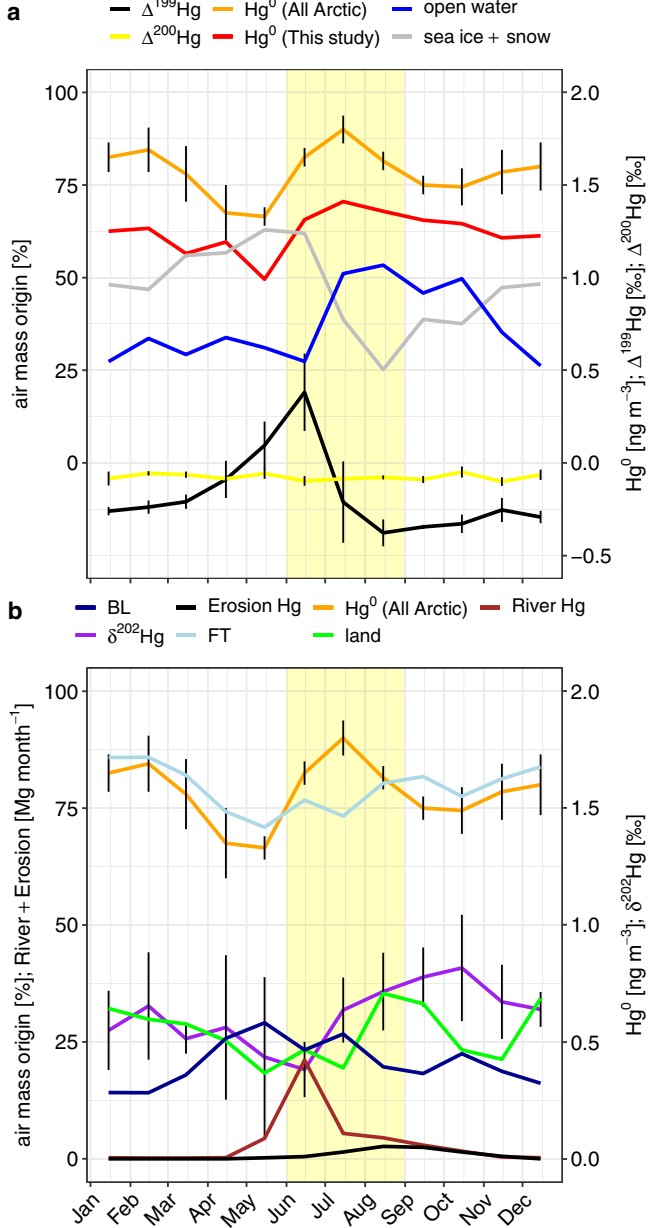

**Fig. 5 | Mean monthly variability in Arctic atmospheric Hg and air mass origins.**
**a** Mean monthly atmospheric $Hg^0$ at Zeppelin (ZEP), Alert (ALT) and Amderma (Russia) stations from 2000 to 2009 (orange line), and mean monthly $Hg^0$ (red line), $\Delta^{199}Hg$ (black line) and $\Delta^{200}Hg$ (yellow line) for ZEP,Villum (VRS) and ALT from this study for 2018–2019. Mean monthly air mass origins in the boundary layer (BL), in %, for 10-day HYSPLIT air mass trajectories ending at ZEP, VRS and ALT are shown for sea ice + snow-covered land (gray line) and open water (blue line). **b** Mean monthly $\delta^{202}Hg$ (purple line), air mass origins over land (%, green line), in the free troposphere (FT, %, lightblue line), boundary layer (BL, %, darkblue line), and monthly pan-Arctic River and coastal erosion Hg inputs (Mg month$^{-1}$). The vertical yellow shaded bar indicates the broad summertime $Hg^0$ maximum. Error bars represent the 2 SD uncertainty for $\Delta^{199}Hg$ and SD uncertainty for $Hg^0$ concentrations.

seasonal air mass provenance maps for boundary layer air as we present here in Fig. 6.

## Arctic $\Delta^{199}Hg$ variability

The $\Delta^{199}Hg$ signature is known to orginate from $Hg^{II}$ photoreduction in water[59] and in snow, in particular during and after AMDEs[41]. Pronounced seasonal $Hg^0$ $\Delta^{199}Hg$ variability can be seen in Fig. 2b relative

to the remote NH median $\Delta^{199}Hg$ of −0.20‰ (IQR −0.13 to −0.25, horizontal red shaded bar), with increased $\Delta^{199}Hg$ from April to June in both years, and decreased $\Delta^{199}Hg$ in August and September. Figure 2b also shows that at the height of the summertime atmospheric $Hg^0$ maximum, $\Delta^{199}Hg$ plunges to its minimal with negative values down to −0.5‰. Sherman et al.[41] previously observed strong odd-Hg isotope MIF during post-AMDE snow $Hg^0$ re-emissions, with $Hg^0$ $\Delta^{199}Hg$ fractionated by +3.3‰ relative to snow $Hg^{II}$. Such large MIF during photoreduction in snow suggests that post-AMDE $Hg^0$ emission is capable of generating enough $Hg^0$ with positive $\Delta^{199}Hg$ to reach the $\Delta^{199}Hg$ values of up to 1.4‰ observed in weekly $Hg^0$ during the AMDE spring months. Sherman et al.[41] also observed that as snow re-emits $Hg^0$ with positive $\Delta^{199}Hg$, the residual snow $Hg^{II}$ fraction progressively attains $\Delta^{199}Hg$ as low as −5.1‰. We observe low snow $\Delta^{199}Hg$ down to −2.4‰ in Ny-Ålesund in 2011 and 2019, and so did others in Toolik, AK (USA) and in ALT[43,56]. Snow $\Delta^{199}Hg/\Delta^{199}Hg$ regression slopes are near 1.0 in our Ny-Ålesund snow data (Fig. 4c), as well as in the Utqiagvik, Toolik and ALT data indicating that a common photochemical mechanism is involved.

Atmospheric reactive $Hg^{II}$ compounds (i.e., aerosol $Hg^{II}$ and gaseous $Hg^{II}$ compounds) at ZEP and VRS during the springtime AMDEs show negative odd-MIF, similar to snow $Hg^{II}$, with a $\Delta^{199}Hg/\Delta^{201}Hg$ ratio of 1.00 ± 0.02‰ (1 SD, Fig. 4c). Reactive $Hg^{II}$ $\Delta^{199}Hg$ decreases progressively from April to June, and the minima down to −2.15‰; coincide exactly, in the same week of sampling, with the maximum $\Delta^{199}Hg$ up to 1.32‰ of $Hg^0$. The similar magnitude and $\Delta^{199}Hg/\Delta^{201}Hg$ slope of odd-MIF in atmospheric reactive $Hg^{II}$ compared to snow $Hg^{II}$ suggests that the photoreduction mechanism and associated odd-Hg magnetic isotope effect that is active in snow[41] also operates in the boundary layer, most likely in aerosols or blowing snow and ice crystals over sea ice and land. The synchronous, yet opposite $\Delta^{199}Hg$ signs in atmospheric both $Hg^0$ and $Hg^{II}$ strongly suggest these signals to be produced in the boundary layer over sea ice and over land and not in surface AO waters. Further, the repetitive seasonal $\Delta^{199}Hg$ variability, detected simultaneously at all three sites, suggests that the snow $Hg^0$ re-emission with positive $\Delta^{199}Hg$ is imprinted regionally on the entire boundary layer $Hg^0$ pool, and possibly on the lower free troposphere.

At the height of the summertime $Hg^0$ maximum in July, $\Delta^{199}Hg$ sharply drops from its positive peak value in June to a negative minimum in July and August (Fig. 2b). This behavior is expected from AMDE Hg re-emissions that undergo strong odd-MIF[41], where early snow $Hg^0$ re-emissions in May and June carry positive $\Delta^{199}Hg$, leading to a residual snow $Hg^{II}$ pool with progressively more negative $\Delta^{199}Hg$ as low as −2 to −5‰. Sherman et al. used a Rayleigh fractionation model to show that beyond 60% of snow $Hg^{II}$ re-emission, the final fractions of re-emitted $Hg^0$ also attain negative $\Delta^{199}Hg$, which potentially explains the observations we make in July and August. Alternatively, we will see below from the air mass back-trajectory analysis that July and August $Hg^0$ emissions are predominantly from regional marine waters. Since there is no evidence for MIF during marine $Hg^{II}$ photoreduction globally[60], it is also possible that the late summer shift to negative $Hg^0$ $\Delta^{199}Hg$ is inherited from AMDE snowmelt runoff, which carries negative $\Delta^{199}Hg$ to marine waters. To support this, we detected isotopic signatures consistent with AMDE runoff in pHg from the surface ocean near the North East Greenland shelf in mid-August, with median $\Delta^{199}Hg$ of −0.20‰ (IQR, −0.16 to −0.27, Fig. 1), which is well below the global marine $\Delta^{199}Hg$ baseline for total Hg of 0.06‰ or for marine sediments of 0.08‰[60].

Finally, we note that the two 3-day integrated $Hg^0$ isotope observations at Utqiagvik, from mid-June 2008[41], have similar $\delta^{202}Hg$ and $\Delta^{200}Hg$ as observed in this study, but distinctly lower $\Delta^{199}Hg$ of −0.1 to −0.2‰ than the positive June 2018–2019 $\Delta^{199}Hg$ observed at ZEP, VRS and ALT. HYSPLIT back trajectories for the short mid-June 2008 observations (Supplementary Fig. 2) indicate mostly open water air mass provenance in the Bering and Chukchi Seas. Longer-term

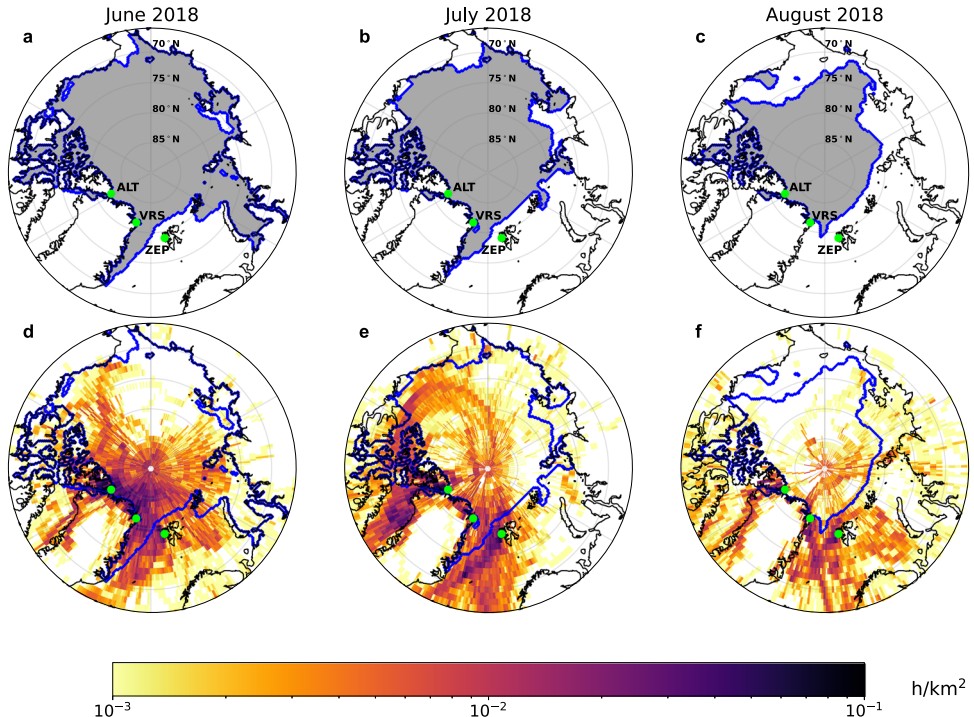

**Fig. 6 | Sea ice extent and origin of air masses arriving at Zeppelin (ZEP), Villum (VRS) and Alert (ALT) research stations from June to August 2018.** Sea ice extent (all panels, blue line) and air mass residence time maps (lower panels, hours per km²) for combined 10-day HYSPLIT back trajectories from ZEP, VRS and ALT. Residence time was calculated only for trajectories within the atmospheric boundary layer. The maps display sea ice extent and air mass residence time for **a**, **d** the 2018 June Δ¹⁹⁹Hg maximum, **b**, **e** July summertime Hg⁰ maximum, and **c**, **f** August Δ¹⁹⁹Hg minimum that were detected at all sites (Figs. 2 and 5). Trajectories in June 2018 and July 2018 originate for 62% and 39% over sea ice and snow-covered coastal land, with an increasing contribution of open water from the Fram Strait in July 2018 and August 2018 (52%). Sea ice extent data from June to August 2018 is obtained from the NSIDC NASA DAAC: National Snow and Ice Data Center[82].

atmospheric Hg⁰ isotope observations need to be made at Utqiagvik to see if sea ice and coastal snow-covered areas lead to the high Δ¹⁹⁹Hg observed further east.

## Arctic δ²⁰²Hg variability

The ZEP, VRS and ALT Hg⁰ δ²⁰²Hg time-series show a large variability, from −0.4 to 1.45‰, but also substantial overlap with the median NH background of 0.43‰ (IQR 0.09 to 0.77; Fig. 2d). Overall, δ²⁰²Hg drops during springtime to reach a minimum of 0.38 ± 0.12‰ that coincides with the Δ¹⁹⁹Hg maximum in June (Fig. 5). δ²⁰²Hg then progressively increases during summer to reach its maximum of 0.82 ± 0.23‰ in October. These observations are again compatible with the Rayleigh fractionation behavior of AMDE Hg re-emissions from snow that are initially, in May and June, enriched in the light Hg isotopes, and then become progressively heavier as the residual snow (on sea ice and on land) HgᴵᴵHg pool becomes enriched in the heavy Hg isotopes. In addition, plant and soil uptake of Hg⁰ during the summertime growth season preferentially removes light Hg isotopes, leading to potentially higher δ²⁰²Hg in residual atmospheric Hg⁰ [61,62]. The overall positive Hg⁰ δ²⁰²Hg contrasts with Arctic terrestrial Hg δ²⁰²Hg. Siberian tundra vegetation and river dissolved dHg runoff shows negative δ²⁰²Hg of −1.0‰ (median, IQR −0.2 to −2.8) and −2.6‰ (median, IQR −2.3 to −4.0) respectively. Similar negative δ²⁰²Hg was observed recently for N-American river run-off in the Mackenzie delta (−1.1 ± 0.6‰ in dHg, −1.5 ± 0.4‰ in pHg, Fig. 4b)[63]. The Siberian and N-American terrestrial Hg inputs can also be observed in marine surface water pHg, which has low δ²⁰²Hg of −1.7 ± 0.4‰ in the transpolar drift current in the central AO and −2.3 ± 0.2‰ in the shallow Barents Sea off Scandinavia (Figs. 1, 4b and Supplementary Fig. 3).

The spreading of less dense river runoff over denser AO waters in estuaries and over the large AO shelf provides, in principle, the conditions where terrestrial Hgᴵᴵ can be (photo-)chemically and biologically reduced and emitted to AO marine boundary layer air. The large amount of terrestrial Hgᴵᴵ (80%) that needs to be photoreduced and emitted from AO surface waters in models[30,34] should lead to minor further enrichment of emitted Hg⁰ in the light Hg isotopes to values below −1.1‰, and possibly lower. The ZEP, VRS and ALT Hg⁰ δ²⁰²Hg data however, do not show evidence of strong light isotope enrichment during the summertime Hg⁰ maximum (Figs. 2 and 4b). Rather, the observed June-August Hg⁰ δ²⁰²Hg of 0.58 ± 0.17‰ is more similar to δ²⁰²Hg observations of Hgᴵᴵ in Arctic coastal snow associated with AMDEs of 0.43‰ (median, IQR 0.08 to 0.75, n = 58)[41,43,56] (this study). Similar to Δ¹⁹⁹Hg, the atmospheric Hg⁰ δ²⁰²Hg observations are therefore compatible with a dominant contribution of AMDE deposited Hg, but not terrestrial Hg.

## Arctic Δ²⁰⁰Hg variability

Δ²⁰⁰Hg, and the related Δ²⁰⁴Hg signature, are thought to be produced by photochemical Hg redox reactions at or above the tropopause[64], and no relevant Hg transformations at the Earth's surface have thus far been shown to further result in even-Hg MIF[40]. Hg redox transformations in the upper atmosphere lead to Hg⁰ and Hgᴵᴵ pools with distinctly different median Δ²⁰⁰Hg⁰ of −0.05‰ (IQR −0.08 to −0.03) and Δ²⁰⁰Hgᴵᴵ of 0.14‰ (IQR 0.09 to 0.18[60]). Δ²⁰⁰Hg is therefore considered a conservative tracer for atmospheric Hg deposition pathways[61], and Δ²⁰⁰Hg of terrestrial surfaces and water bodies reflect the relative proportions of Hg⁰ and Hgᴵᴵ deposition. For example, recent work has suggested that global soils, runoff and lake sediments, with Δ²⁰⁰Hg of 0.00‰, reflect ~75% vegetation and soil Hg⁰ uptake and ~25% Hgᴵᴵ wet and dry deposition[61,65]. Similarly, pelagic marine waters, sediment and biota with average slightly positive Δ²⁰⁰Hg of 0.04‰ reflect 50% of ocean Hg⁰ uptake and 50% Hgᴵᴵ wet and dry deposition[60].

In Fig. 3, we compare the persistently negative $Hg^0$ $\Delta^{200}Hg$ of −0.08‰ at ZEP, VRS and ALT to observations of $\Delta^{200}Hg$ in Russian and N-American tundra vegetation[66], to Yenisei and Mackenzie River dHg and pHg, to surface AO pHg, and to atmospheric $Hg^{II}$ and snow THg at ZEP, VRS and ALT. We observe that Russian vegetation has a positive median $\Delta^{200}Hg$ of 0.08‰, which is directly reflected in the constant $\Delta^{200}Hg$ of 0.04‰ in dHg of the Yenisei River, the largest Arctic river in terms of discharge and Hg input to the AO[33,34]. Mackenzie River dHg and pHg also carry median positive $\Delta^{200}Hg$ of 0.06‰, and 0.04‰ respectively (Fig. 3 and Supplementary Fig. 4). Figure 3 illustrates that atmospheric $Hg^0$ $\Delta^{200}Hg$ at all three sites is significantly different from both Russian tundra vegetation and Yenisei and Mackenzie river runoff $\Delta^{200}Hg$ (Wilcoxon two sample t test, $p < 0.05$). The seasonal $Hg^0$ $\Delta^{200}Hg$ flatlines of −0.08‰; at ZEP, VRS and ALT (Fig. 2c) therefore corroborate the back-trajectory, $\delta^{202}Hg$, and $\Delta^{199}Hg$ observations and suggest that there is not a large Russian or N-American terrestrial Hg contribution (due to the positive $\Delta^{200}Hg$ for Russian vegetation, Yenisei and Mackenzie River Hg), via AO $Hg^0$ emission, to the summertime $Hg^0$ maximum at these sites.

Interestingly, Alaskan tundra vegetation (Fig. 3, Vegetation, Toolik) has a lower median $\Delta^{200}Hg$ of −0.03‰ than Eurasian vegetation (Lichen/Moss, Siberia) and runoff (River, Yenisei, Mackenzie). This is likely due to latitude bias in the data, with Alaskan tundra data from >67°N, and most of the Siberian tundra vegetation data and >90% of the Yenisei and Mackenzie River watersheds from <67°N. $Hg^{II}$ wet deposition, with its elevated $\Delta^{200}Hg$ of 0.14‰, is known to increase towards the mid-latitudes, explaining the relatively positive $\Delta^{200}Hg$ of 0.04–0.06‰ in dHg of the Yenisei and Mackenzie Rivers. Surface AO pHg, which reflects internally and externally sorbed $Hg^{II}$ on dead and living marine particulate matter, has a median $\Delta^{200}Hg$ of 0.01‰ (IQR −0.02 to 0.05, Fig. 1, Supplementary Fig. 3), which potentially reflects multiple source contributions to $Hg^{II}$ in marine particles: incoming Atlantic and Pacific waters, terrestrial Hg, and atmospheric $Hg^0$ and $Hg^{II}$. We observe relatively little geographic surface AO pHg $\Delta^{200}Hg$ variation across Fram Strait, the Barents Sea, and the central basin, except two samples (out of 32, Fig. 1, Supplementary Fig. 3) in the transpolar drift current, which merits further study in the future.

During AMDEs, $Hg^0$ is thought to be efficiently and near-quantitatively oxidized into reactive $Hg^{II}$, deposited to snow, and then partially photoreduced and re-emitted as $Hg^0$. Membrane-collected reactive $Hg^{II}$ at ZEP and VRS during the springtime AMDE seasons have median $\Delta^{200}Hg$ of −0.03‰ (IQR −0.07 to 0.02), and snow $Hg^{II}$ at Ny-Ålesund and across the Arctic have median $\Delta^{200}Hg$ of −0.11‰ (IQR −0.11 to −0.04) and −0.05‰ (IQR −0.08 to −0.01) respectively (Fig. 3). Hg in snowmelt at Utqiagvik also showed $\Delta^{200}Hg$ of −0.08 and −0.09‰ ($n = 2$[23]). Within the analytical uncertainty, the negative $Hg^0$ $\Delta^{200}Hg$ signatures are therefore conserved during the chain of AMDEs into snow $Hg^{II}$, and snowmelt $Hg^{II}$ which has also been observed by others[41,42]. The $\Delta^{200}Hg$ of −0.03‰ in reactive $Hg^{II}$ forms sampled at ZEP and VRS are significantly higher than the simultaneously sampled $Hg^0$ with $\Delta^{200}Hg$ of −0.08‰ ($p < 0.05$). The absolute $\Delta^{200}Hg$ difference between $Hg^{II}$ and $Hg^0$ is small however, and at the limit of atmospheric $Hg^{II}$ isotope analytical uncertainty. In addition, while membranes collect more $Hg^{II}$ than denuder-based methods in the Arctic[67], there is a possibility that over a 1-week sampling period some gaseous or particulate $Hg^{II}$ was lost from the membranes, leading to unpredictable minor bias in $Hg^{II}$ $\Delta^{200}Hg$ (but not $Hg^0$ $\Delta^{200}Hg$).

## Cause of the summertime $Hg^0$ maximum

Air mass back-trajectory analysis (Figs. 5 and 6) shows that the summertime boundary layer air masses, with their distinct $\delta^{202}Hg$, $\Delta^{199}Hg$ and $\Delta^{200}Hg$ signatures typical of AMDE re-emission dynamics, have their origins over sea ice and snow-covered coastal land (62%) in the N-Greenland and Ellesmere Island region in June. In July, boundary layer air masses shift progressively from snow-covered sea ice and land

(39%) to open waters (51%) of Baffin Bay and Fram Strait. In August, during the descent from the $Hg^0$ peak, these proportions reach 25% for snow-covered ice and land and 53% for open waters. Boundary layer trajectories over land are 26% on average from June to August. At no time do boundary layer trajectories reach ice-free Siberian shelf waters where most terrestrial Hg is discharged. These findings therefore contradict previous suggestions that terrestrial Hg inputs from Russian rivers and coastal erosion to the AO, followed by 80% oceanic emissions to the atmosphere, make an important contribution to the summertime $Hg^0$ maximum observed throughout the Arctic[25,30,34]. While there is broad overlap in the timing between estimated river discharge and $Hg^0$ $\Delta^{199}Hg$ maxima (Fig. 2b), $\Delta^{199}Hg$ increases with the onset of AMDEs about 1 month before substantial river Hg discharge. In addition, the Siberian and Beaufort Sea shelves remain ice-covered until mid-June, 2 months after the first $\Delta^{199}Hg$ increases, and during the steepest increase in atmospheric $Hg^0$ (Supplementary Figs. 5 and 6). This suggests that while atmospheric $Hg^0$ concentrations (and $Hg^0$ $\Delta^{199}Hg$) and river Hg discharge partially co-vary, they are not causally related.

The dominant interaction of July and August air masses with open marine waters raises the question if marine $Hg^0$ emissions contributing to the July $Hg^0$ maximum represent recent AMDE Hg runoff from snowmelt over ice and coastal land (including glaciers) to marine waters or non-AMDE related marine Hg derived from background atmospheric deposition to the same regional marine waters throughout the year. Marine total Hg concentrations over the North East Greenland shelf and in Baffin Bay are indeed elevated in August, reaching up to 4 pM in surface waters due to meltwater inputs[68,69]. The August pHg isotope data over the North East Greenland shelf also support a meltwater Hg source to the atmosphere, because of its relatively elevated $\delta^{202}Hg$ of −0.40‰ and low $\Delta^{199}Hg$ of −0.20‰ (Fig. 1), which contrast with the terrestrial signatures observed in the transpolar drift current further north.

Previous Arctic Hg budgets of coastal erosion Hg flux are variable, 16–47 Mg y$^{-1}$, depending on assumed, and highly variable, glacial sediment Hg concentrations. Here we propose a new, lower estimate of 9 Mg y$^{-1}$ (Supplementary Table 2), taking benefit of improved erosional mass and carbon budgets and deep mineral soil and coastal glacial sediment Hg/C ratios. We estimate seasonal coastal erosion Hg fluxes (Fig. 2b, and Supplementary Data File) by scaling the annual flux of 9 Mg y$^{-1}$ to monthly estimates of erosional C and N fluxes[35]. Coastal erosion Hg inputs are too small and arrive too late, peaking late August and September, to make a contribution to the summertime atmospheric $Hg^0$ maximum. On the contrary, we document how river Hg inputs imprint their low $\delta^{202}Hg$, and positive $\Delta^{199}Hg$ and $\Delta^{200}Hg$ on surface AO pHg in August 2015 (Fig. 1). This suggests that terrestrial Hg from rivers and coastal erosion likely impacts AO marine ecosystems but not the atmosphere. We therefore conclude, based on air mass origins and Hg isotope signatures, that the summertime atmospheric $Hg^0$ peak is most likely sustained by re-emission of Hg deposited during the previous spring season to the cryosphere. This re-emission takes place directly from the cryosphere but also from regional open marine waters that receive meltwater Hg inputs.

Recent studies have quantified a large permafrost soil Hg pool, approximately 72,000 Mg in the upper 30 cm[36], which results in an important river and coastal erosion Hg flux to the AO. Ongoing Arctic warming has fueled concerns about enhanced mobilization of permafrost Hg to surface AO (0–200 m) ecosystems and to the global atmosphere, which contain relatively smaller amounts of 270 and 4000 Mg of Hg, respectively.[6,69] Models incorporating this terrestrial Hg flux to the AO suggest its partial reduction in AO surface waters and emission to the atmosphere[25,30,34,37], where it would then sustain the summertime $Hg^0$ maximum. The model predictions of AO emission of terrestrial Hg inputs hinge on the key assumption that approximately 80% of the terrestrial Hg inputs are reduced with the remaining 20%

depositing to shelf sediments. Our findings, based on the Hg isotope fingerprints of Arctic atmospheric $Hg^0$, do not show evidence of a terrestrial origin for summertime $Hg^0$. We speculate that terrestrial Hg is mostly deposited to AO shelf sediments, without dramatically evading to the global atmosphere. The large AO shelf area supports a rich and diverse ecosystem, and enhanced deposition of terrestrial Hg to shelf sediments may lead to enhanced microbial Hg methylation and increased MeHg exposure to the benthic and pelagic food webs and ultimately humans. More work is needed to assess the impact of an increased terrestrial Hg load on coastal Arctic ecosystems.

# Methods

## Study area

Atmospheric Hg isotope observations were made at Zeppelin observatory (ZEP, Svalbard), Villum Research Station Nord (VRS, Greenland), and Alert station (ALT, Canada) (Fig. 1). All three stations are situated far from major air pollution point sources. ZEP is located on the top of Mt. Zeppelin, Svalbard (78.90°N, 11.89°E, 474 meters above sea level (m a.s.l.)), just outside the community of Ny-Ålesund. A steep downhill slope faces north towards the research village of Ny-Ålesund, a small settlement with 35 to 185 inhabitants at 2 km from the sampling site. VRS is located in the north-eastern corner of Greenland on the north-south oriented peninsula Princesse Ingeborgs, which is a 20 × 15 km Arctic lowland plain. Measurements were performed at the Air Observatory at Villum (81.6°N, 16.67°W, 24 m a.s.l.), which is located on the Danish military base Station Nord. The atmospheric observatory at Villum is located 2 km to the south of Station Nord and is upwind >95% of the time from local pollution sources at the military base. The ALT location (82.5°N, 62.3°W, 200 m a.s.l.) is at the Dr. Neil Trivett Global Atmosphere Watch Observatory at Alert, Nunavut, Canada and is located about 8 km south of the Lincoln Sea.

## Atmospheric $Hg^0$ sampling and processing

Activated carbon traps impregnated with sulfur (HGR-AC, Calgon Carbon Corp., Pittsburg, PA, USA) were used to collect atmospheric $Hg^0$ isotopes at ZEP, VRS and ALT stations. Weekly and limited bi-weekly sampling was conducted from March, 2018 to June, 2019 (ZEP, VRS) and from May, 2018 to March, 2019 (ALT). The sampling train consisted of a 47 mm, 0.45 μm porosity polyethersulfone (PES) cation exchange membrane (Merck Millipore) in a 47 mm Savillex PFA Teflon filter holder, connected by 6 mm FEP Teflon tubing to HGR-AC traps (200 mg activated carbon powder, in 10 cm long, 4 mm internal, 7 mm external diameter Pyrex glass tubes), a ball flow meter (Fisher scientific), a digital volume meter (Siargo Ltd.), and a membrane vacuum pump (KNF). The PES membranes collect gaseous and aerosol $Hg^{II}$ species, and the HGR-AC traps collect $Hg^0$. At ALT, a calibrated MKS mass flow controller and Gast carbon vane pump were used. The sampling flow of 0.5–1.0 L min⁻¹ was regularly checked with a calibration unit (Bios Defender) and considered stable throughout the campaigns. After weekly field sampling, HGR-AC traps were sealed with silicone stoppers, packed in a double-zipper bag, and stored frozen on site. Samples were transported frozen to Toulouse, France, at the end of campaigns for Hg isotope analysis. HGR-AC traps were combusted in a double-stage tube furnace in pure $O_2$ at 80 mL min⁻¹, and purged and pre-concentrated into 8 mL of 40% v/v $HNO_3$/HCl in a 2:1 ratio (Sun et al.[70]). Purging impingers were rinsed with a total volume of 8 mL of MQ water, diluting the sample to 20% v/v acid. The final trapping solutions were kept in a refrigerator at 2–4 °C until Hg isotope analysis.

During the campaigns, automated Tekran® 2537 instruments (Tekran® Inc., Canada) measured $Hg^0$ continuously, at 5 to 15 min resolution, at all three stations. Tekran® 2537 models pump and pre-concentrate ambient $Hg^0$ over gold traps, and then $Hg^0$ is thermally desorbed and detected by cold vapor atomic fluorescence spectrometry (CVAFS). The air inlets for all automated and manual air Hg sampling were installed 3–5 m above ground in close proximity to one

another, facing downward and toward the predominant wind directions. Identical operating procedures were used at the sites and strict quality control criteria are documented elsewhere (Berg et al.[51]; Boyd Pernov et al.[71]; Steffen et al.[20]). HGR-AC trap $Hg^0$ recoveries were estimated from Tekran® 2537 and trap solution Hg concentrations and were 90 ± 21% at ZEP and VRS (mean ± 2 SD).

## Atmospheric reactive $Hg^{II}$ sampling and processing

Atmospheric $Hg^{II}$ was collected weekly during spring in 2018 and 2019 at ZEP and VRS, when $Hg^{II}$ levels during AMDE events are high enough for Hg isotope analysis. The sampling train, validated elsewhere (Fu et al.[64]), consisted of a 90 mm, 0.45 μm porosity polyethersulfone (PES) cation exchange membrane (Merck Millipore) in a 90 mm Savillex PFA Teflon filter holder, connected by 6 mm FEP Teflon tubing to a ball flow meter set at 4.0 L min⁻¹ (Fisher scientific), a digital volume meter (Siargo Ltd.), and a membrane vacuum pump (KNF). PES membranes were sealed in petri-dishes, packed in double-zipper bags, stored in a freezer (−20 °C) on site, and transported frozen to France for analysis. In the laboratory, PES membranes were leached in 16 mL of 2.5% v/v $HNO_3$/HCl in a 2:1 ratio, in acid-cleaned 50 mm diameter PFA Teflon beakers (Savillex) on a hot plate (Analab) at 120 °C. The $Hg^{II}$ concentration in the PES leachates was determined using a Brooks Rand Model III CV-AFS with a custom-made purge and trap system. The method detection limit (MDL) was 5 pg $Hg^{72}$. When Hg concentrations were high enough, leachate solutions were combined and submitted to a purge and trap pre-concentration: leachates were diluted with MQ water to 0.5 L volume in a 1.0 L pre-cleaned Pyrex bottle with GL45 Savillex ¼" two-port cap. $SnCl_2$ (2.5 mL of 3 wt% $SnCl_2$ in 1 N HCl) was added to the bottle and purged with Hg-free argon for 6 h at 400 mL min⁻¹ into a 8 mL, 40% v/v $HNO_3$/HCl (2:1 ratio) trap[42]. Final trap solutions were diluted with MQ water to 20% v/v acid and kept refrigerated at 2–4 °C until Hg isotope analysis.

## Snow, river water and marine particles sampling

Snow samples were collected in 2011 and 2019 from 40 cm deep pits dug outside of Ny-Ålesund close to the base of the Zeppelin mountain. All the samples were kept at −20 °C in the dark onsite, and transported to France frozen, where Hg-free BrCl was added upon thawing to convert all Hg species to labile $Hg^{II}$ forms.

The Yenisei River was sampled at Igarka (67.4°N, 86.4°E), 300 km from the river mouth, using a boat or via holes drilled in the ice cover. Hg samples were filtered in the field using pre-burnt 47 mm, 2.0 μm porosity quartz filters (QMA Millipore) and a 47 mm Savillex PFA Teflon filter holder into acid-cleaned 500 mL FEP Teflon bottles, acidified to 0.36 M with bi-distilled HCl, stored at 4 °C in the dark until transport to France. Sample aliquots were analyzed by CVAFS and dissolved Hg (dHg) concentrations published elsewhere[34]. Remaining samples were stored cold at 4 °C until BrCl addition and pre-concentration by the same purge and trap procedure mentioned above for $Hg^{II}$ leachates and adapting purge and trap bottle volume where needed (1, 5, 20 L). Pre-concentration recoveries were found to be 85 ± 20% (2 SD) for the rivers and snow samples.

Marine particles were sampled on QMA filters with in situ pumps (McLane) in the Barents Sea and the central Arctic Ocean were sampled during the GEOTRACES TransArc II (GN04) cruise (17th August to 15th October 2015) and in Fram Strait during the 2016 GEOTRACES GRIFF (GN05) cruise (18th July to 6th September 2016) aboard the FS Polarstern. Sampling details and concentration of particulate Hg (pHg) are given in refs. [69], [73]. QMA filters were combusted in a double-stage tube furnace in pure $O_2$ at 80 mL min⁻¹, and purged and pre-concentrated into 8 mL of 40% v/v $HNO_3$/HCl in a 2:1 ratio[70].

## Hg isotope analysis

Hg isotope ratios were measured on a Neptune Plus multi-collector inductively coupled plasma mass spectrometer (MC-ICPMS, Thermo-

Finnigan) at the Observatoire Midi-Pyrénées, Toulouse, France, and at the University of Toronto, Canada, following the methods described in previous studies[74,75]. In Toulouse, we used a CETAC ASX-520 auto-sampler and HGX-200 CV system coupled with the MC-ICPMS, equipped with a $10^{12}$ Ω amplifier on the $^{198}$Hg isotope in order to improve isotope ratio precision in the 50 mV range. Sample and standard signals at 0.3 to 1.0 ng g$^{-1}$ Hg concentrations were generally 150–500 mV on the $^{202}$Hg isotope, at a sample introduction flow rate of 0.5 mL min$^{-1}$. Thallium was not used as an internal standard, and the 203 and 205 masses were occasionally monitored to survey Hg-hydride interferences (i.e., $^{202}$Hg$^{1}$H, and $^{202}$Hg$^{1}$H$^{1}$H), which were found to be negligible when using standard H-cones. In Toronto, Hg was introduced to the MC-ICPMS by reducing Hg$^{II}$ in liquid samples with 3% SnCl$_2$ to Hg$^0$ vapor, which was separated using a gas-liquid separator. Instrumental mass bias of MC-ICPMS was corrected by standard-sample-standard bracketing using NIST3133 Hg at matching concentrations. In addition to sample-standard bracketing, mass bias at University of Toronto was also corrected using Tl as an internal standard (NIST SRM 997) that was introduced using an Aridus II desolvating nebulizer. Hg isotopic composition is reported in delta notation (δ) in units of per mil (‰) referenced to the bracketed NIST 3133 Hg standard[76]:

$$\delta^{xxx}Hg(‰) = \left[ \frac{\left(\frac{^{xxx}Hg}{^{198}Hg}\right)_{sample}}{\left(\frac{^{xxx}Hg}{^{198}Hg}\right)_{NIST3133}} - 1 \right] \times 10^3 \qquad (1)$$

where xxx represent Hg isotopes 199, 200, 201, 202, 204. Mass-independent isotope fractionation (MIF) is expressed in "capital delta (Δ)" notation (‰), which is the difference between the measured values of $\delta^{199}$Hg, $\delta^{200}$Hg, $\delta^{201}$Hg, and $\delta^{204}$Hg and those predicted from $\delta^{202}$Hg using the kinetic mass-dependent fractionation law:

$$\triangle^{xxx}Hg(‰) = \delta^{xxx}Hg - (\beta^{xxx} \times \delta^{202}Hg) \qquad (2)$$

where $\beta^{xxx}$ is 0.2520, 0.5024, 0.7520, 1.493 for the 199, 200, 201, and 204 Hg isotopes respectively. Analytical uncertainty of isotopic analysis was assessed by repeated measurement of the samples, of in-house standards UM-Almaden, ETH Fluka and JT Baker, and of procedural standards NIST SRM 1632d and 1632e (see Supplementary data file). The results obtained were in agreement with published values[61,62,77,78]. The 2 SD uncertainty reported for samples is the largest of the 2 SD's of sample replicates, procedural standards, or in-house standards.

### Back-trajectory analysis

The Hybrid Single-Particle Lagrangian Integrated Trajectory model (HYSPLIT, v. 4.2.0; https://www.arl.noaa.gov/hysplit/) developed by NOAA[79]. was driven with 3 hourly meteorological input data from the Global Data Analysis System (GDAS; 1° latitude-longitude 360 by 181 grid) to identify the potential Hg source regions. The model was run in backward mode for 240 h every 2 h throughout the sampling periods at ZEP, VRS and ALT. In total, ~84 backward trajectories were calculated for each sampling period. An analysis of the spatial (horizontal and vertical) residence time of the air mass history has been accomplished. The back-trajectory model results have been analyzed with respect to the air residence time above five surface types (land without snow cover, open water, permanent ice/snow, sea ice, and land based snow). The surface type maps "land without snow cover", "open water" and "permanent ice/snow" were extracted from Climate Change Initiative (CCI) land cover data, the "sea ice" coverage from the AMSR-2 sea ice concentration product (sea ice coverage >50% sea ice concentration) and the "land based snow" from MODIS/Terra and MODIS/Aqua data sets. Data from MYD10C1 (Aqua) were used to fill gaps in MYD10C1 (Terra) (see Supplementary Table 1 for details). We used ESRI ArcGIS

Pro (v. 10.6) to provide the percent surface exposure of particles along trajectories for each surface type. In addition, the percent of particles along trajectories that reside within the boundary layer (BL) and in the free troposphere (FT) has been calculated.

## Data availability

The research data that support the findings of this study are available at https://doi.org/10.3929/ethz-b-000549236. Meteorological and field data used in HYSPLIT simulations from 2018 and 2019 are available from NOAA (https://www.ready.noaa.gov/archives.php). Sea ice concentration data from April to December 2018 and 2019, respectively, displayed in the Supplement, were obtained from https://www.meereisportal.de (grant: REKLIM-2013-04).

## Code availability

The residence time maps in Fig. 6 were created using the python script available at https://github.com/ShaddyAhmed/HYSPLIT-Hg. The HYSPLIT model and READY website can be accessed here: https://www.ready.noaa.gov.

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

## Acknowledgements

This study was funded by the French Polar Institute IPEV (Program 1207 MESSI), the EC H2020 ERA-PLANET (grant No. 689443) iCUPE program, the EC H2020 INTERACT project (grant No 730938), the Swiss National Science Foundation (SNSF) project P400P2_180796, the NSF project 1700711, the Chantier Arctique Français (Pollution in the Arctic System) project, the AXA Research Fund and a Make Our Planet Great Again (MOPGA) postdoctoral scholarship from the French Ministry of Science and Education. We thank the following persons for assistance in the field and laboratory: Jerome Chmeleff from GET, Alban Thollot from IGE, Ove Hermansen and Are Bäcklund from NILU, Bjarne Jensen, Christel Christoffersen, and Keld Mortensen from AU, Christelle Guesnon from NPI, Melody Fraser and Kevin Rawlins at Alert and all personnel from IPEV and Alfred Wegener Institute involved in this study. We thank Frank Wania for providing activated carbon sorbent for sampling. The authors also acknowledge the NOAA Air Resources Laboratory (ARL) for the provision of the HYSPLIT transport and dispersion model and READY website (https://www.ready.noaa.gov) used in this publication.

## Author contributions

J.E.S., A.D., B.B. A.S., J.L.T., T.P., N.T., R.T., H.S. and L.E.H.B. planned the study. B.F.A., S.O., G.S., O.M., H.S., K.A.P., J.B.P., K.N., N.T., C.N., R.T., A.M., L.E.H.B., and J.E.S. performed fieldwork and/or managed monitoring activities. B.F.A., S.O., L.L., N.S., D.L., B.B., L.E.H.B., M.V.P. and J.E.S. performed laboratory analyses. S.O., S.A. and S.S. performed back-trajectory modeling and data visualization. J.E.S. and S.O. led paper writing. All authors contributed to data interpretation and writing.

## Competing interests

The authors declare no competing interests.
