## [Peer Review File · Nature Communications]

Mercury isotope evidence for Arctic summertime re-emission of mercury from the cryosphereREVIEWER COMMENTS

Reviewer #1 (Remarks to the Author):

Review: Mercury isotope evidence for Arctic summertime re-emission of mercury from the cryosphere

The article presents a robust dataset that details Hg cycling in the arctic. They found average Hg(0) atmospheric concentrations to be lower than the northern hemisphere and found that Arctic atmospheric Hg now matches the observed global decline. They also highlight in-depth analysis of isotopic mercury variability throughout the year+ study. Overall, the article is well-written and presents important data for arctic Hg cycling particularly in a warming climate setting. The article would be stronger if a 'Conclusions' sections is added after the 'Implications' section.

Line items:

Line 45: add a comma after 'marine'

Line 57: add a citation for the long-range atmospheric transport of Hg.

Lines 150 and 175: one heading is italicized and one is not, please make consistent

Lines 165 to 174: this paragraph needs a bit more discussion like in the other sections, what have other studies found, similar or different?

Line 301: not sure why the one phrase is highlighted

Line 331: I like the addition of an 'Implications' section, but I would consider adding also a 'Conclusions' section, as well. The authors present a large amount of data with nice figures, a summary at the end would be helpful for the reader as well.

Lines 406 and 407: add a break between the heading and paragraph like the rest of the sections

Line 413: define frozen – -4C or ultra-cold? And elsewhere in the methods change out ‘frozen’ with the actual temperature

Line 430: stored ‘cold’.. 4C?

Figure 1: add a comma after the location for Villum

Reviewer #2 (Remarks to the Author):

This paper found that the Hg(0) in the summertime Arctic atmosphere is mostly from re-emission of Hg deposited to ice and snow during AMDE, but not from re-emission of terrestrially discharged Hg. The main evidence is Hg isotopes and air mass back trajectory. This study is a large undertaking with a full suite of atmospheric Hg isotope data from multiple sites. The paper is clearly written and the data quality is high. While I in general agree with the authors' conclusion, as it is known that the re-emission from ice and snow is high, I have some concerns about the interpretation of Hg isotope data, particularly those in summer. I think the Hg isotope data since August may indicate re-emission of Hg(0) by photoreduction in seawater and is not necessarily from Hg deposited to the cryosphere during spring. Please see the comments below.

1.L203-208: I fully agree that the strongly positive MIF of Hg(0) in June is best explained by photochemical re-emission from ice and snow. However, I have two questions. First, why did not March, April and May show similarly positive MIF in Hg(0)? Does this suggest the re-emission in Spring is much lower than in Summer? In the ZEP site, the MIF of Hg(II) shows a sharp decrease from -0.2 to -2 from April to June, but Hg(0) shows only a small increase. This seems to be opposite to what is expected from a mass balance. Second, regarding the interpretation on the negative shift of MIF in July and August, the authors used a Rayleigh fractionation model which predicts that the MIF of Hg(0) would follow the trend of MIF of remaining Hg(II) in snow. However, if a Rayleigh fractionation model is followed, then from April to June when the Hg(II) shows a sharp decrease of MIF, one would expect to see a decrease of MIF in Hg(0) as well (if the Hg(0) is produced by photoreduction in blowing snow as proposed by the authors), which is in fact opposite to the observed MIF trend in Hg(0). Moreover, the change in MIF caused by a Rayleigh fractionation should be progressive, not as abrupt as the switch from positive to negative from July to August. I think a more intuitive interpretation for the negative MIF of Hg(0) in August is that the re-emission is not dominated by ice or snow anymore, but by seawater. Photoreduction in seawater would normally produce negative MIF for Hg(0), right? This could also explain the abruptness of the change in the direction of MIF from July to August. We should also notice that there was a large variability in MIF between different sites in June and July, but the variability is much smaller since August. This also points to a change in the source of Hg(0). Photoreduction in snow

and ice can produce large variation in MIF because snow and ice are quite heterogeneous. But seawater is a more mixed pool of Hg(II), which is consistent with the less variability of MIF since August. Also note the negative shift of Hg(0) MIF in August corresponds well with an increase in air mass origin from open water (and thus a decrease from sea ice) (Figure 5), supporting a change in the source of Hg(0). Would this interpretation compromise the authors' main conclusion? Anyway, I think the temporal variation of MIF in Hg(0) needs more consideration.

2. For the δ^{202} , the authors argue the positive δ^{202} in Hg(0) is inconsistent with the negative δ^{202} of terrestrial Hg. That is true. But in summer time, the δ^{202} of Hg(0) can be modified by terrestrial plants. The absorption of Hg(0) by plants would cause a large MDF and leave the residual atmospheric Hg(0) positive δ^{202} values. This process cannot be ignored. The δ^{202} can be also modified by other processes (e.g., non-photo redox reactions in seawater) that have relatively small effects on MIF.

3.L315: Is the δ^{202} positive or negative?

4. There are too many abbreviations. It is OK to use abbreviation for sites, but even "boundary layer" and "free troposphere" are using abbreviation. Is it really necessary? They are not common abbreviations, so it is harder for readers to keep reminding themselves what BL and FT are. I suggest to use full names when their abbreviations are uncommon.

Reviewer #1:

Comment 1: The article presents a robust dataset that details Hg cycling in the arctic. They found average Hg(0) atmospheric concentrations to be lower than the northern hemisphere and found that Arctic atmospheric Hg now matches the observed global decline. They also highlight in-depth analysis of isotopic mercury variability throughout the year+ study. Overall, the article is well-written and presents important data for arctic Hg cycling particularly in a warming climate setting. The article would be stronger if a 'Conclusions' sections is added after the 'Implications' section.

Response 1: Following the editor's advice above, we suggest to group the closing paragraph called 'Implications' with the previous section that concludes on the 'cause of the summertime Hg⁰ peak'.

Specific comments:

Line 45: add a comma after 'marine'

Done

Line 57: add a citation for the long-range atmospheric transport of Hg

We have added the reference [Dastoor et al. \(2022\): "Arctic mercury cycling NatRev EarthEnv"](https://www.nature.com/articles/s43017-022-00269-w)
<https://www.nature.com/articles/s43017-022-00269-w>

Lines 150 and 175: one heading is italicized and one is not, please make consistent

The section headings were changed to bold only throughout.

Lines 165 to 174: this paragraph needs a bit more discussion like in the other sections, what have other studies found, similar or different?

Few Hg studies have used dedicated back trajectory analysis to evaluate the origin of the summertime Hg⁰ maximum. Most Hg (and ozone and halogen) back trajectory work has been done in the context of springtime AMDEs, indicating central Arctic Ocean origins over sea ice. We have added the following discussion to address the reviewer's question (L174 – 180):

“Previous studies on the arctic summertime Hg⁰ peak (Steffen et al., 2005; Fisher et al., 2012, 2013; Zhang et al., 2015; Sonke et al., 2018) did not assess air mass origins. Seasonal air mass provenance was determined using 10-day HYSPLIT back trajectories in a pan-Arctic aerosol study (Freud et al., 2017), finding similar lack of origins over Siberian coastal waters for ALT and VRS. A study on longterm Hg⁰ observations at VRS used five-day back trajectories, finding no strong seasonal correlations between Hg⁰ and time spent over sea ice, open water, land or snow (Skov et al., 2020). The study did not, however, produce seasonal air mass provenance maps for boundary layer air as we present here in Figure 6.”

This figure is Figure 12 in Freud et al. (2017): *The seasonal areal coverage of 240 h long back-trajectories for each of the sites. The top maps compare winter and summer in blue and red, respectively, while the bottom maps compare the spring and autumn. Only grid cells with a trajectory-crossing probability greater than 0.5 % are shown.*

Line 301: not sure why the one phrase is highlighted

The phrase should not have been highlighted. We have changed that.

Line 331: I like the addition of an ‘Implications’ section, but I would consider adding also a ‘Conclusions’ section, as well. The authors present a large amount of data with nice figures, a summary at the end would be helpful for the reader as well.

Please see reply above: at present the last two paragraphs summarize main findings and put them into context. We would prefer not adding another paragraph, but we are open to further advice.

Lines 406 and 407: add a break between the heading and paragraph like the rest of the sections

Done

Line 413: define frozen – -4C or ultra-cold? And elsewhere in the methods change out ‘frozen’ with the actual temperature

We define “frozen” in L 401: [...] and stored frozen (-20 °C) on site. Samples were transported frozen to Toulouse, [...]. This definition is valid for the entire text.

Line 430: stored ‘cold’.. 4C?

Yes, stored at 4°C. We have changed that in L440: [...] stored at 4°C in the dark [...]

Figure 1: add a comma after the location for Villum

Done

Reviewer #2:

Comment 1: This paper found that the Hg(0) in the summertime Arctic atmosphere is mostly from re-emission of Hg deposited to ice and snow during AMDE, but not from re-emission of terrestrially discharged Hg. The main evidence is Hg isotopes and air mass back trajectory. This study is a large undertake with a full suite of atmospheric Hg isotope data from multiple sites. The paper is clearly written and the data quality is high. While I in general agree with the authors' conclusion, as it is known that the re-emission from ice and snow is high, I have some concerns about the interpretation of Hg isotope data, particularly those in summer. I think the Hg isotope data since August may indicate re-emission of Hg(0) by photoreduction in seawater and is not necessarily from Hg deposited to the cryosphere during spring. Please see the comments below.

Response 1: We have extended the discussion about the summertime D¹⁹⁹Hg isotope signal, and provide detailed answers below to the sea water re-emission question raised. In short, we agree with the reviewer on a late summer regional marine Hg emission source, but it appears our writing was not clear on this topic, despite our conclusion on L341 that “This re-emission takes place directly from the cryosphere but also from regional open marine waters that receive meltwater Hg inputs”.

Comment 2: L203-208: I fully agree that the strongly positive MIF of Hg(0) in June is best explained by photochemical re-emission from ice and snow. However, I have two questions. First, why did not March, April and May show similarly positive MIF in Hg(0)? Does this suggest the re-emission in Spring is much lower than in Summer?

Response 2: Correct, net re-emission is lower in Spring (March, April, May) than in summer, which is fully reflected in the atmospheric Hg⁰ concentration seasonality, which drops in Spring (net deposition) and rises dramatically only in June. Δ¹⁹⁹ MIF rises slightly above the winter baseline by April and May, suggesting some re-emission in these months, but the bulk of re-emissions clearly occur in summer and are reflected in the Δ¹⁹⁹ seasonality.

Comment 3: In the ZEP site, the MIF of Hg(II) shows a sharp decrease from -0.2 to -2 from April to June, but Hg(0) shows only a small increase. This seems to be opposite to what is expected from a mass balance.

Response 3: There are two answers to this: To some extent we would argue that these opposite MIF observations in the Hg⁰ and Hg^{II} pools do reflect mass balance. The -1.8 per mil negative shift for Hg^{II} is larger because the Hg^{II} pool size is much smaller than the Hg⁰ pool size: i.e. Hg^{II} concentrations are 0.04 ng m⁻³ in spring at ZEP, while Hg⁰ concentrations are 1.5 ng m⁻³. Atmospheric photoreduction of Hg^{II}, producing Hg⁰ with positive MIF, will strongly shift the residual Hg^{II} pool to negative MIF, but not make a dent (small increase MIF) in the larger Hg⁰ pool. However, the snow Hg^{II} pool should be included in such a mass balance, because snow Hg^{II} photoreduction and re-emission drive the Hg⁰ MIF signature more than atmospheric Hg^{II} photoreduction.

In reality, our Hg^{II} and Hg⁰ MIF data only represent a snapshot of the very dynamic chemical and physical processes that affect atmospheric Hg speciation at the cryosphere-atmosphere interface, on its way to the ZEP observatory. This means that during the week of sampling, air with Hg^{II} and Hg⁰ arriving 3

at ZEP is a mix of AMDE Hg^0 re-emissions, inputs of free tropospheric Hg^0 and Hg^{II} , and partial loss of Hg^{II} by deposition to snow, ice, water and land. From that perspective, our Hg concentration and isotope data do not permit a full mass balance (which we do not make). Efforts are under way to add Hg isotopes to atmospheric chemistry and transport models, which can then be used for a more quantitative ‘mass balance’ approach by comparing to our data.

Comment 4: Second, regarding the interpretation on the negative shift of MIF in July and August, the authors used a Rayleigh fractionation model which predicts that the MIF of $\text{Hg}(0)$ would follow the trend of MIF of remaining $\text{Hg}(\text{II})$ in snow. However, if a Rayleigh fractionation model is followed, then from April to June when the $\text{Hg}(\text{II})$ shows a sharp decrease of MIF, one would expect to see a decrease of MIF in $\text{Hg}(0)$ as well (if the $\text{Hg}(0)$ is produced by photoreduction in blowing snow as proposed by the authors), which is in fact opposite to the observed MIF trend in $\text{Hg}(0)$. Moreover, the change in MIF caused by a Rayleigh fractionation should be progressive, not as abrupt as the switch from positive to negative from July to August. I think a more intuitive interpretation for the negative MIF of $\text{Hg}(0)$ in August is that the re-emission is not dominated by ice or snow anymore, but by seawater. Photoreduction in seawater would normally produce negative MIF for $\text{Hg}(0)$, right?

Response 4: We use the Rayleigh analogy to provide one possible explanation of the Δ^{199} variability; it is not known whether true closed system Rayleigh conditions are met at any point for the Springtime polar dome, or polar boundary layer. Most likely not, because snow Hg^0 emissions mix dynamically into moving air masses. We show in Figure 5 that at any time 75% of free troposphere background air (with FT Hg^0 with likely Δ^{199} of -0.2 per mil) is mixed into the $\text{Hg}^0 \Delta^{199}$ signals we detect at the three stations. This means that the Δ^{199} peaks of the emitted Hg^0 are in reality 4x larger than the diluted signal we measure. The variability between stations in spring is also dictated by the free troposphere mixing effects. Later in summer when Δ^{199} is negative and less variable, we suspect that the polar dome is breached and that we are simply looking at mixing of arctic and mid-latitude air. Below we further comment that we agree with the reviewer on a late summer marine Hg emission source. We do not think that marine Hg^{II} photoreduction leads to MIF (there is no widespread evidence for this), but we agree with the reviewer that there is room for an alternative interpretation (alternative from Rayleigh) for the negative Δ^{199} during late summer. We added the following on lines 214 – 219: “Alternatively, we will see below from the air mass back trajectory analysis that July and August Hg^0 emissions are predominantly from regional marine waters. Since there is no evidence for MIF during marine Hg^{II} photoreduction globally (Jiskra et al., 2021), it is also possible that the late summer shift to negative $\text{Hg}^0 \Delta^{199}\text{Hg}$ is inherited from AMDE snowmelt runoff, carrying negative $\Delta^{199}\text{Hg}$, to marine waters.”

This could also explain the abruptness of the change in the direction of MIF from July to August. We should also notice that there was a large variability in MIF between different sites in June and July, but the variability is much smaller since August. This also points to a change in the source of $\text{Hg}(0)$. Photoreduction in snow and ice can produce large variation in MIF because snow and ice are quite heterogeneous. But seawater is a more mixed pool of $\text{Hg}(\text{II})$, which is consistent with the less variability of MIF since August. Also note the negative shift of $\text{Hg}(0)$ MIF in August corresponds well with an increase in air mass origin from open water (and thus a decrease from sea ice) (Figure 5), supporting a change in the source of $\text{Hg}(0)$. Would this interpretation compromise the authors' main conclusion?

Response 4 continued: We appreciate these detailed questions by the reviewer; it reminds us of the long debates amongst the authors to understand the factors that drive the large MIF variability at the three sites. We address the question of a marine origin to the late summer atmospheric Hg^0 peak on lines 321 – 324 of the MS: “The dominant interaction of July and August air masses with open marine waters

raises the question if marine Hg^0 emissions contributing to the July Hg^0 maximum represent recent AMDE Hg runoff from snowmelt over ice and coastal land (including glaciers) to marine waters, or non-AMDE related marine Hg derived from background atmospheric deposition to the same marine waters throughout the year.”

We therefore agree with the reviewer on a dominant marine origin of the late summer Hg^0 peak, and we ask, in the MS, the question what the origin of that marine Hg is? Does it result from AMDE runoff (from snow melt on ice, and snow melt on coastal land) to the North East Greenland Sea and to Baffin Bay, or from baseline (already present) marine Hg in these waters? We argue in the same paragraph (325 – 329) that “Marine total Hg concentrations over the North East Greenland shelf and in Baffin Bay are indeed elevated in August, reaching up to 4 pM in surface waters due to meltwater inputs (Wang et al., 2018; Petrova et al., 2020). The August pHg isotope data over the North East Greenland shelf support a meltwater Hg source to the atmosphere, because of its relatively elevated $\delta^{202}\text{Hg}$ of -0.40‰ and low $\Delta^{199}\text{Hg}$ of -0.20‰ (Figure 1), which contrast with the terrestrial signatures observed in the transpolar drift current further north.”

And we conclude on L341 “This re-emission takes place directly from the cryosphere but also from regional open marine waters that receive meltwater Hg inputs.”

We therefore agree with the reviewer, and the consideration of regional marine Hg^0 emission of meltwater Hg^{II} inputs does not change our main conclusion that the large Siberian river and erosional Hg inputs do not drive the summertime pan-arctic Hg peak.

Comment 5: Anyway, I think the temporal variation of MIF in $\text{Hg}(0)$ needs more consideration.

Response 5: Agreed; please see above responses and text modifications

Comment 6: For the d_{202} , the authors argue the positive d_{202} in $\text{Hg}(0)$ is inconsistent with the negative d_{202} of terrestrial Hg. That is true. But in summer time, the d_{202} of $\text{Hg}(0)$ can be modified by terrestrial plants. The absorption of $\text{Hg}(0)$ by plants would cause a large MDF and leave the residual atmospheric $\text{Hg}(0)$ positive d_{202} values. This process cannot be ignored. The d_{202} can be also modified by other processes (e.g., non-photo redox reactions in seawater) that have relatively small effects on MIF.

Response 6: Agreed, though it is unclear how large such an effect would be (perhaps a few tenths of a per mil). At the arctic tundra station of Toolik, late summer (and year-round) Hg^0 $\delta^{202}\text{Hg}$ is 0.7‰ which is within the northern hemisphere $\delta^{202}\text{Hg}$ variability that we cite: IQR 0.09‰ – 0.77‰.

We have added a phrase on L236 to acknowledge the possibility of a vegetation effect on $\delta^{202}\text{Hg}$: “In addition, plant and soil uptake of Hg^0 during the summertime growth season is enriched in the light Hg isotopes, leading to potentially higher $\delta^{202}\text{Hg}$ in residual atmospheric Hg^0 (Demers et al., 2013; Enrico et al., 2016).”

Comment 7: L315: Is the d_{202} positive or negative?

Response 7: The δ^{202} is negative. The isotopic $\delta^{202}\text{Hg}$ signature from the Northeast Greenland Shelf was -0.4 ‰. No changes were made in the text.

Comment 8: There are too many abbreviations. It is OK to use abbreviation for sites, but even "boundary layer" and "free troposphere" are using abbreviation. Is it really necessary? They are not common abbreviations, so it is harder for readers to keep reminding themselves what BL and FT are. I suggest to use full names when their abbreviations are uncommon.

Response 8: We follow the Reviewer's suggestion and now use full names for "BL", "FT" in the text. We also deleted the abbreviation for magnetic isotope effect (MIE) in L199. To our knowledge, these were the only uncommon abbreviations in the manuscript. Station names (ALT, VRS, ZEP), Atmospheric Mercury Depletion Events (AMDEs) or Arctic Ocean (AO) ideally remain abbreviated.

References:

- Dastoor, A., Angot, H., Bieser, J., Christensen, J.H., Douglas, T.A., Heimbürger-Boavida, L.-E., Jiskra, M., Mason, R.P., McLagan, D.S., Obrist, D., Outridge, P.M., Petrova, M.V., Ryjkov, A., St. Pierre, K.A., Schartup, A.T., Soerensen, A.L., Toyota, K., Travnikov, O., Wilson, S.J., Zdanowicz, C., 2022. Arctic mercury cycling. *Nat Rev Earth Environ* 3, 270–286. <https://doi.org/10.1038/s43017-022-00269-w>
- Demers, J. D., Blum, J. D., and Zak, D. R.: Mercury isotopes in a forested ecosystem: Implications for air-surface exchange dynamics and the global mercury cycle, 27, 222–238, <https://doi.org/10.1002/gbc.20021>, 2013.
- Enrico, M., Le Roux, G., Maruszczak, N., Heimbürger, L.-E., Claustres, A., Fu, X., Sun, R., and Sonke, J. E.: Atmospheric mercury transfer to peat bogs dominated by gaseous elemental mercury dry deposition., <https://doi.org/10.1021/acs.est.5b06058>, 2016.
- Fisher, J. A., Jacob, D. J., Soerensen, A. L., Amos, H. M., Steffen, A., and Sunderland, E. M.: Riverine source of Arctic Ocean mercury inferred from atmospheric observations, 5, 499–504, <https://doi.org/10.1038/ngeo1478>, 2012.
- Fisher, J. A., Jacob, D. J., Soerensen, A. L., Amos, H. M., Corbitt, E. S., Streets, D. G., Wang, Q., Yantosca, R. M., and Sunderland, E. M.: Factors driving mercury variability in the Arctic atmosphere and ocean over the past 30 years, 27, 1226–1235, <https://doi.org/10.1002/2013gb004689>, 2013.
- Freud, E., Krejci, R., Tunved, P., Leaitch, R., Nguyen, Q. T., Massling, A., Skov, H., and Barrie, L.: Pan-Arctic aerosol number size distributions: seasonality and transport patterns, 17, 8101–8128, <https://doi.org/10.5194/acp-17-8101-2017>, 2017.
- Jiskra, M., Heimbürger-Boavida, L.-E., Desgranges, M.-M., Petrova, M. V., Dufour, A., Ferreira-Araujo, B., Masbou, J., Chmeleff, J., Thyssen, M., Point, D., and Sonke, J. E.: Mercury stable isotopes constrain atmospheric sources to the ocean, *Nature*, 597, 678–682, <https://doi.org/10.1038/s41586-021-03859-8>, 2021.
- Skov, H., Hjorth, J., Nordstrøm, C., Jensen, B., Christoffersen, C., Bech Poulsen, M., Baldtzer Liisberg, J., Beddows, D., Dall'Osto, M., and Christensen, J. H.: Variability in gaseous elemental mercury at Villum Research Station, Station Nord, in North Greenland from 1999 to 2017, 20, 13253–13265, <https://doi.org/10.5194/acp-20-13253-2020>, 2020.
- Sonke, J. E., Teisserenc, R., Heimbürger-Boavida, L.-E., Petrova, M. V., Maruszczak, N., Le Dantec, T., Chupakov, A. V., Li, C., Thackray, C. P., Sunderland, E. M., Tananaev, N., and Pokrovsky, O. S.: Eurasian river spring flood observations support net Arctic Ocean mercury export to the atmosphere and Atlantic Ocean, 115, E11586–E11594, <https://doi.org/10.1073/pnas.1811957115>, 2018.
- Steffen, A., Schroeder, W., Macdonald, R., Poissant, L., and Konoplev, A.: Mercury in the Arctic atmosphere: An analysis of eight years of measurements of GEM at Alert (Canada) and a comparison with observations at Amderma (Russia) and Kuujuarapik (Canada), 342, 185–198, <https://doi.org/10.1016/j.scitotenv.2004.12.048>, 2005.
- Zhang, Y., Jacob, D. J., Dutkiewicz, S., Amos, H. M., Long, M. S., and Sunderland, E. M.: Biogeochemical drivers of the fate of riverine mercury discharged to the global and Arctic oceans, 29, 854–864, <https://doi.org/10.1002/2015gb005124>, 2015.

REVIEWERS' COMMENTS

Reviewer #1 (Remarks to the Author):

The authors have done a great job addressing reviewer comments. I recommend this work for publication.

Reviewer #3 (Remarks to the Author):

I read the comments from the previous reviewer 2 and fully agree with them. I think the authors have incorporated most of the comments from reviewer 2 and addressed them properly. A key point in the original comment is that the MIF in summer can be interpreted as re-emission from seawater and not necessarily from snow/ice. The authors acknowledged and incorporated this alternative interpretation, and further argued that the source of the marine Hg is most likely from AMDE meltwater rather than terrestrial input. Thus this alternative interpretation is still consistent with the main conclusion. Overall, Hg isotopes provide a unique evidence for the origin of Hg re-emission in the Arctic, which help distinguishing the contributions of different Hg sources to marine ecosystem and atmospheric environment in the Arctic.

I just have a minor comment for Response 2: So yes, the Hg(0) concentration suggests low re-emission in spring. But is this observation consistent with previous studies? There should be a comment on this.

Point-by-point response to the second round of comments by the Reviewers on “Mercury isotope evidence for Arctic summertime re-emission of mercury from the cryosphere” (NCOMMS-22-06630B)

Reviewer #1:

Comment 1 The authors have done a great job addressing reviewer comments. I recommend this work for publication.

Response 1: Thank you very much. We appreciate your comments.

Reviewer #3:

Comment 1: I read the comments from the previous reviewer 2 and fully agree with them. I think the authors have incorporated most of the comments from reviewer 2 and addressed them properly. A key point in the original comment is that the MIF in summer can be interpreted as re-emission from seawater and not necessarily from snow/ice. The authors acknowledged and incorporated this alternative interpretation, and further argued that the source of the marine Hg is most likely from AMDE meltwater rather than terrestrial input. Thus this alternative interpretation is still consistent with the main conclusion. Overall, Hg isotopes provide a unique evidence for the origin of Hg re-emission in the Arctic, which help distinguishing the contributions of different Hg sources to marine ecosystem and atmospheric environment in the Arctic.

I just have a minor comment for Response 2: So yes, the Hg(0) concentration suggests low re-emission in spring. But is this observation consistent with previous studies? There should be a comment on this.

Response 1: Yes the observation of low atmospheric Hg⁰ concentration in spring is consistent with previous studies. It has been made by all arctic Hg monitoring stations. We refer to the relevant literature (refs 25-27) in lines 73-75: “The springtime atmospheric Hg⁰ depletion (mean of 1.35 ng m⁻³ for April-May, 2000-2009 period) observed at different monitoring stations across the Arctic including Alert, Villum, Zeppelin, Utqiagvik, and Amderma is generally followed by a summertime Hg⁰ maximum of 1.80 ng m⁻³ (July mean²⁵⁻²⁷)”. In our previous answer to comment 2 by Reviewer 2 we wrote:

“...net re-emission is lower in spring (March, April, May) than in summer, which is fully reflected in the atmospheric Hg⁰ concentration seasonality, which drops in spring (net deposition) and rises dramatically only in June. Δ¹⁹⁹ MIF rises slightly above the winter baseline by April and May, suggesting some re-emission in these months, but the bulk of re-emissions clearly occur in summer and are reflected in the Δ¹⁹⁹ seasonality.”

Here we add to this previous reply, and focus on the re-emission part of the reviewer’s question: “But is this observation (of lower Hg⁰ re-emission in spring) consistent with previous studies?”. Our suggestion is supported by GRAHAM model simulations of the Arctic Hg cycle that indicate that Arctic Ocean evasion (66°N – 90°N) is clearly lower in May (5 ng m⁻² d⁻¹) compared to June (16 ng m⁻² d⁻¹) and July (20 ng m⁻² d⁻¹) (Dastoor and Durnford, 2014, Figure 2 excerpt below). However, direct Hg⁰ exchange flux observations over all seasons from the Arctic Ocean, including sea ice area are lacking (Dastoor et al., 2022). Such observations will be helpful in the future to confirm model-estimated Arctic Ocean Hg⁰ evasion and Hg isotope constraints from our study. In order to support our suggestion, we have added the following sentence to the manuscript, with the reference (L76): “This unique Arctic Hg⁰ seasonality suggests net Hg deposition in spring, followed by net Hg emission during summer (Dastoor and Durnford, 2014).”

Figure R1. Shown, at Alert and Ny-Ålesund sites for 2008, are observed (red) and simulated (blue) concentrations of atmospheric Hg^0 , simulated revolatilization of Hg^0 from snowpacks (magenta), and simulated ocean evasion (green) of Hg^0 from adjacent open ocean waters. Summertime maxima in atmospheric Hg^0 concentrations are indicated by gray shading. A six-hour running mean centered on the stated date/time was applied to Hg^0 concentrations. Evasion values are daily totals. Revolatilization values are scaled (divided by four) daily totals.

References:

- Dastoor, A.P., Durnford, D.A., 2014. Arctic Ocean: Is It a Sink or a Source of Atmospheric Mercury? *Environ. Sci. Technol.* 48, 1707–1717. <https://doi.org/10.1021/es404473e>
- Dastoor, A., Angot, H., Bieser, J., Christensen, J.H., Douglas, T.A., Heimbürger-Boavida, L.-E., Jiskra, M., Mason, R.P., McLagan, D.S., Obrist, D., Outridge, P.M., Petrova, M.V., Ryjkov, A., St. Pierre, K.A., Schartup, A.T., Soerensen, A.L., Toyota, K., Travnikov, O., Wilson, S.J., Zdanowicz, C., 2022. Arctic mercury cycling. *Nat Rev Earth Environ* 3, 270–286. <https://doi.org/10.1038/s43017-022-00269-w>.